# Continuous Object Representation Networks: Novel View Synthesis without Target View Supervision

**Nicolai Häni**      **Selim Engin**      **Jun-Jee Chao**      **Volkan Isler**
{haeni001, engin003, chao0107, isler}@umn.edu
University of Minnesota

## Abstract

Novel View Synthesis (NVS) is concerned with synthesizing views under camera viewpoint transformations from one or multiple input images. NVS requires explicit reasoning about 3D object structure and unseen parts of the scene to synthesize convincing results. As a result, current approaches typically rely on supervised training with either ground truth 3D models or multiple target images. We propose Continuous Object Representation Networks (CORN), a conditional architecture that encodes an input image's geometry and appearance that map to a 3D consistent scene representation. We can train CORN with only two source images per object by combining our model with a neural renderer. A key feature of CORN is that it requires no ground truth 3D models or target view supervision. Regardless, CORN performs well on challenging tasks such as novel view synthesis and single-view 3D reconstruction and achieves performance comparable to state-of-the-art approaches that use direct supervision. For up-to-date information, data, and code, please see our project page [1].

## 1 Introduction

In 1971, Shephard and Metzler [44] introduced the concept of mental rotation, the ability to rotate 3D objects mentally and link the model to its projection. Novel View Synthesis (NVS) research seeks to replicate this capability in machines by generating images of a scene from previously unseen viewpoints, unlocking applications in image editing, animation, or robotic manipulation. View synthesis is a challenging problem, as it requires understanding the 3D scene structure, reason on image semantics, and the ability to render the internal representation into a target viewpoint. A common approach for NVS is to use a large collection of views to reconstruct a 3D mesh [10, 43]. Recent methods have made progress in learning 3D object representations, such as voxel grids [60, 46, 52, 33, 32], point clouds [1, 61, 56], or meshes [54, 12, 48, 55]. However, the discrete nature of these representations limit the achievable resolution and induce significant memory overhead. Continuous representations [36, 25, 42, 47, 58, 6, 24, 27] address these challenges. However, proposed methods require either 3D ground truth or multi-view supervision, limiting these approaches' applicability to domains where data is available.

We introduce Continuous Object Representation Networks (CORNs), a neural object representation that enforces multi-view consistency in geometry and appearance with natural generalization across scenes, learned from as few as two images per object. The key idea of CORNs is to extract local and global features from the input images and represent the scene implicitly as a continuous, differentiable function that maps local and global features to 3D world coordinates. We optimize CORNs from only two source views using transformation chains and 3D feature consistency as self-supervision, requiring $50\times$ fewer data during training than the current state-of-the-art models.

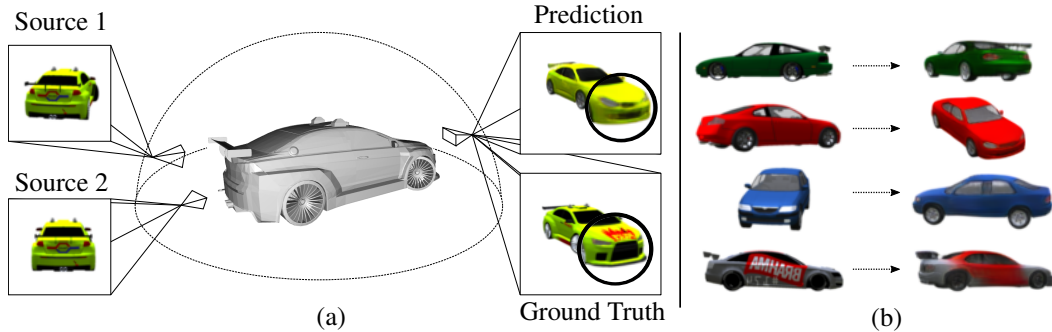

Figure 1: Our model learns to synthesize novel views using only two source images per object during training (a). For this instance, even though both source images are from the back of the car, our model can reconstruct unseen areas with a reasonable detail level. During training, the target view prediction is *not* directly supervised with ground truth. Instead, it is transformed into the second source image while maintaining the consistency of the learned representation. During inference (b), our model predicts novel views from a single input image. It can accommodate drastically different source and target poses.

The conditional formulation of CORNs, combined with a differentiable neural renderer, enforces multi-view consistency and allows for the fast inference of novel views from a single image during test time, without additional optimization of latent variables. We evaluate CORNs on various challenging 3D computer vision problems, including novel view synthesis, 3D model reconstruction, and out of domain view synthesis.

To summarize, our approach makes the following contributions:

- A continuous, conditional novel view synthesis model, CORNs based on a novel representation that captures the scene's appearance and geometry at arbitrary spatial resolution. CORNs are end-to-end trainable and uses only two images per object during training time, without any 3D space or 2D target view supervision.

- Despite being self-supervised, CORN performs competitively or even outperforms current state-of-the-art approaches that use dozens of images per object and direct supervision on the target views.

- We demonstrate several applications of our method, including novel view synthesis, single-view 3D reconstruction, and novel view synthesis from out-of-domain samples.

## 2 Related Work

Our goal is to generate novel camera perspectives of static 3D scenes. As such, our work lies at the intersection of novel view synthesis, 3D object reconstruction, and generative modeling. In the following, we review related work in these areas.

**Novel view synthesis.** Novel view synthesis is the problem of generating new camera perspectives of a scene. Key challenges of novel view synthesis are inferring the scene's 3D structure and inpainting occluded and unseen parts. Existing methods differ in their generality, some aim to learn a general model for a class of objects [59, 33, 5, 47], while others learn instance-specific models [46, 26, 28]. Training an instance-specific model generally produces higher quality results, at the cost of lengthy training times for each object instance. For real-world applications, this is often prohibitive. Improving general models is an open problem, and as CORNs generalize naturally across object instances, we focus our literature review on methods that synthesize novel views for a general category of objects.

Traditionally, novel view synthesis uses multi-view geometry [7, 10, 43] to triangulate 3D scene content. Once the 3D scene is reconstructed, novel views can be generated by rendering the resulting 3D mesh. Instead of explicit 3D mesh reconstruction, other approaches have sought to represent 3D knowledge implicitly; by directly regressing pixels in the target image [50, 49, 59], weakly disentangling view pose and appearance [67, 62, 22] or by learning appearance flow [65, 35, 49, 5].

Other prior work proposed to apply the view transformations in latent space [57] or learn a complete latent space of posed objects [51] from which to sample. While these techniques successfully predict novel views under small viewpoint transformations, they do not allow 3D structure extraction. Our proposed method encapsulates both the scenes 3D structure and appearance and can be trained end-to-end via a differentiable renderer.

**Neural scene representations.** Deep-learning-based image rendering has become an active research area, creating a plethora of geometric proxy representations. Broadly, these representations can be categorized on whether they represent 3D geometry implicity or explicitly. Explicit representations include voxels [60, 46, 52, 33, 32], meshes [54, 12, 48, 55] or point clouds [1, 61, 56]. While these discretization based methods have enabled some impressive results, they are memory intensive and limited in the representation of complicated surfaces. To improve upon these shortcomings, recent work focuses on learning neural scene representations. Generative Query Networks (GQN) (GQN) [9, 23, 31] a framework to learn low dimensional embedding vectors that represent both the 3D scenes structure and appearance. While GQNs allow sampling of 2D view samples consistent with its lower-dimensional embedding, they disregard the scenes 3D structure.

Continuous function representations represent 3D space as the level set of a function, parametrized by a memory-efficient neural network, which can be sampled to extract 3D structure. Different function representations have emerged, such as binary occupancy classifiers [6, 24, 27], signed distance functions [36, 25, 42, 47, 58] or volumetric representations [28]. While these techniques are successful at modeling 3D geometry, they often require 3D supervision. When combined with a differentiable renderer, some approaches are supervised with 2D target images instead, relying on large image collections for training. However, it can be difficult for real-world applications to obtain dozens or even hundreds of images of each object we would like to model. In contrast, our proposed method encapsulates scene geometry and appearance from *only two reference images per object* and can be trained end-to-end via a learned neural rendering function through self-supervision.

**Generative models.** Our work builds on recent advances in generating high-quality images with deep neural networks. Especially Generative Adversarial Networks (GAN) [13, 38, 3] and its conditional variants [30, 16, 66] have achieved photo-realistic image generation. Some approaches synthesize new views by incorporating explicit spatial or perspective transformations into the network [15, 17, 60]. Another approach is to treat novel view synthesis as an inverse graphics problem [22, 62, 21, 53, 45, 63]. However, these 2D generative models only learn to parametrize 2D views and their respective transformations and struggle to produce multi-view consistent outputs since the underlying 3D structure cannot be exploited.

## 3 Method

Given a dataset $\mathcal{D} = \{(I_{1,2}^i, K_{1,2}^i, T_{1,2}^i)\}_{i=1}^N$ of N objects, each consisting of a tuple with two images $I_{1,2}^i \in \mathbb{R}^{H \times W \times 3}$ and their respective intrinsic $K_{1,2}^i \in \mathbb{R}^{3 \times 3}$ and extrinsic $T_{1,2}^i \in \mathbb{R}^{3 \times 4}$ camera matrices, our goal is to learn a function $f$ that synthesizes novel views at arbitrary goal camera poses $T_G^i \in \mathbb{R}^{3 \times 4}$ (Fig. 2). We parametrize $f = f_\theta$ as a neural network with parameters $\theta$ that naturally enforces 3D structure and enables generalization of shape and appearance across objects. We are interested in a conditional formulation of $f_\theta$ that requires no additional optimization of latent variables at inference time, and that can be optimized from only two images per object. In the following, we first introduce the three components of our network and then discuss how to optimize with limited data from only two input images per object. For notational simplicity, we drop the superscript denoting the specific object.

### 3.1 Feature encoding

The feature encoder network *e* maps input images to a lower-dimensional feature encoding. Inspired by Xu et al. [58], we hypothesize that combining a global feature encoding with spatial pixel-wise features increases the level of detail of the generated images, which we confirm in Sec. 4. The global encoder predicts a global feature vector $z$ that should represent object characteristics such as geometry and appearance. The spatial feature encoder predicts feature maps at the same resolution as the input image. Sampling from this feature map should represent scene semantics beyond merely RGB color and provide additional details to the 3D scene representation.

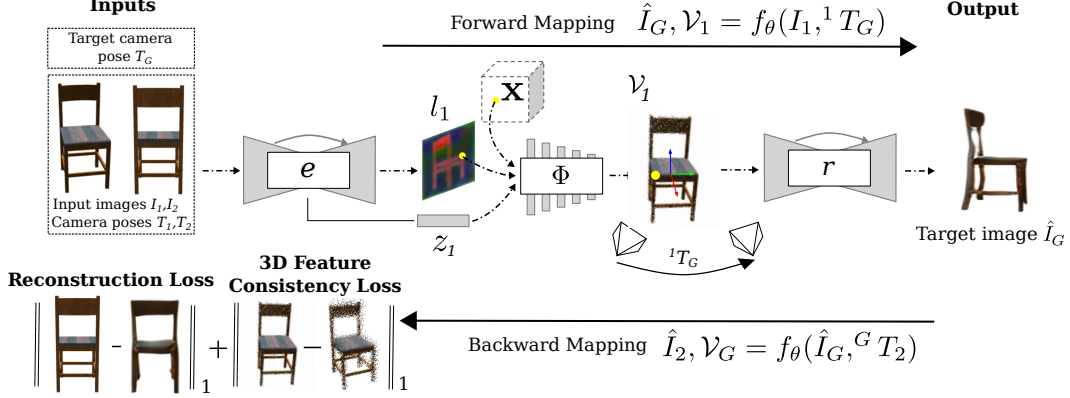

Figure 2: **Our proposed end-to-end model.** CORN takes two source images $I_{1,2}$, together with their respective camera poses $T_{1,2}$, and a target camera pose $T_G$ as input. We aim to learn a function $\hat{I}_G = f_\theta(I_1, {}^1T_G)$ that synthesizes novel views $\hat{I}_G$ at target pose $T_G$. Our approach consists of three parts. The *feature predictor e* embeds the input image in a lower dimensional feature space (visualization by projecting features with PCA). The *neural scene representation*, $\Phi$, maps these features to a 3D consistent neural representation in world coordinates $(x, y, z)$ (the diagram shows RGB for clarity). Finally, a *neural renderer r* renders the scene from arbitrary novel views $T_G$.

**Global feature encoder.** To predict a global feature embedding, we use a ResNet-18 [14] network to extract a 128-dimensional global feature $z \in \mathbb{R}^{128}$. We initialize the ResNet-18 with weights pretrained on the ImageNet dataset and allow further optimization during training.

**Spatial feature network**. The spatial feature network builds on the UNet [40] architecture. It takes the last two-dimensional feature map of the global feature encoder as input, followed by four upsampling and skip-connection layers to extract a 64-dimensional per-pixel feature $l \in \mathbb{R}^{64 \times H \times W}$ of the same size as the input image.

## 3.2 Neural scene representation

Our neural scene representation $\Phi$ maps a spatial location $\mathbf{x} \in \mathbb{R}^3$, the global object descriptor $z$, and local features $l$ to a feature representation of learned scene properties at spatial location $\mathbf{x}$. The feature representation may encode visual information, such as RGB color, but it may also encode higher-order information, such as binary occupancy. In contrast to discrete representations, such as voxel grids or point clouds, which only sparsely sample object properties, $\Phi$ densely models object properties, which can, in theory, be sampled at arbitrary resolutions. In contrast to recent work on representing scenes as continuous functions with a single global object descriptor [36, 47] we combine global and local features. Combining local and global features is similar to recent work by Xu et al. [58] on single-view 3D model reconstruction, which has shown improved performance on modeling fine details.

Our implicit representation $\Phi$ is aware of the 3D structure of objects, as the input to $\Phi$ contains world coordinates $\mathbf{x}$. We sample $k$ 3D points $\{\mathbf{x}_j\}_{j=1}^k$ uniformly at random from a cubic volume and extract local features by projecting the 3D points to the feature map $l$ using the known camera pose. We follow a perspective pinhole camera model that is fully specified by its extrinsic $E = [R, t] \in \mathbb{R}^{3 \times 4}$ and intrinsic $K \in \mathbb{R}^{3 \times 3}$ camera matrices. The extrinsic camera matrix contains rotation matrix $R \in \mathbb{R}^{3 \times 3}$ and translation vector $t \in \mathbb{R}^3$. Given a 3D coordinate $\mathbf{x}_j$, the projection from world space to the camera frame is given by:

$$\mathbf{u} = [u \quad v \quad 1]^\top = K(R\mathbf{x}_j + t) \tag{1}$$

where $u$ and $v$ are the pixel coordinates in the image plane. We extract local features $l_{uv}$ using bilinear sampling, which is fast to compute and differentiable. The neural representation network $\Phi$ takes concatenated global features, local features, and world coordinates $(z, l_{uv}, \mathbf{x}) \in \mathbb{R}^m$ as input and maps them to a higher dimensional feature $\mathbf{v}_j \in \mathbb{R}^n$ at the given spatial coordinate $\mathbf{x}_j$:

$$\Phi : \mathbb{R}^m \to \mathbb{R}^n, (z, l_{uv}, \mathbf{x}_j) \mapsto \Phi(z, l_{uv}, \mathbf{x}_j) = \mathbf{v}_j \tag{2}$$

We denote the collection of features at sampled points by $\mathcal{V} = \{\mathbf{v}_j\}_{j=1}^k$, and in our experiments use a 64-dimensional feature $\mathbf{v}_j$ for each spatial location. We also predict occupancy probabilities for each 3D point to increase spatial consistency, which we render as binary masks. In Sec. 4, we show that this formulation leads to multi-view consistent novel view synthesis results.

### 3.3 Point encoding

Instead of operating on $(x, y, z)$ world-coordinates directly, we found that learning a higher dimensional embedding of the 3D space improves the overall performance. This is consistent with recent work [39, 29], which show that neural networks are biased towards learning low frequency functions. A higher dimensional encoding of world-coordinates can increase the network's capacity to model high-frequency details. Previous works also showed, that using deterministic trigonometric functions for the point encoding $\gamma$ achieves similar performance as when $\gamma$ is parametrized as an MLP. Following Mildenhall et al. [29] we parametrize $\gamma$ as non-learned composition of trigonometric functions:

$$\gamma(\mathbf{x}) = \big(\sin(2^0\pi\mathbf{x}), \cos(2^0\pi\mathbf{x}), ..., \sin(2^{L-1}\pi\mathbf{x}), \cos(2^{L-1}\pi\mathbf{x})\big). \tag{3}$$

In our experiments, we set $L = 10$ and apply $\gamma$ to each point coordinate individually.

### 3.4 Neural renderer

Given a 3D feature space $\mathcal{V}$ of a scene sampled at 3D points, we introduce a neural rendering algorithm that maps a 3D feature space $\mathcal{V}$ as well as the intrinsic $K$ and extrinsic camera parameters $T$ to a novel view $\hat{I}_G$ target camera pose $T_G$. The sampled feature points are projected to the image plane at the target camera's transformation matrix $T_G$ high-performance renderer based on [56]. The neural renderer projects a 3D feature point $\mathbf{v}_j$ to a region in image space with center $c_j$, and radius $r$, where the features influence on a given pixel $p_{uv}$ is given by it's Euclidean distance $d$ to the region's center:

$$p_{mn} = \begin{cases} \mathcal{N}(\mathbf{v}_j, p_{uv}) = 0 & \text{if } d(c_j, p_{uv}) > r \\ \mathcal{N}(\mathbf{v}_j, p_{uv}) = 1 - \dfrac{d(c_j, p_{uv})}{M} & \text{otherwise} \end{cases} \tag{4}$$

where $r$ and $M$ are controllable hyper-parameters. Although $\mathcal{N}$ is non-differentiable, Wiles et al. [56] approximate derivatives using the sub-derivative. The projected points are accumulated in a z-buffer and sorted according to the distance from the new camera before accumulating into a projected feature map $\bar{\mathcal{V}} \in \mathbb{R}^{64 \times H \times W}$ using alpha over-compositing. For additional technical details, please refer to the excellent description in [56].

To render the high dimensional projected features into an RGB image, we use a refinement network $r$. The refinement network renders color values, infers missing features, and reasons for the image's occluded regions. The refinement network is built on a UNet [40] architecture with four down/upsampling blocks and skip connections and spectral normalization [39] following each convolution layer to regularize training.

### 3.5 Training details

To discover a meaningful 3D scene representation without 3D or 2D target image supervision, we assume without loss of generality that for an object, there exists a unique 3D object representation in a canonical view frame. We define the objects' frontal view (0° azimuth and 0° elevation) as the canonical view. To learn such a canonical 3D representation, we offer two key insights: 1) If at least two source images per object are available, we can use transformation chains as supervision, and 2) for the same object, the sampled 3D feature space $\mathcal{V}$ has to be multi-view consistent. In the following, we introduce two loss terms that enforce these insights.

**Transformation chain loss.** Given two source images, $I_1$ and $I_2$ of an object, we learn a multi-view consistent 3D feature space through transformation chains. We define a transformation chain as:

$$\hat{I}_2 = f_\theta(f_\theta(I_1, {}^1T_G,)^G T_2) \tag{5}$$

where we first transform source image $I_1$ with relative transformation ${}^1T_G$ from camera pose $T_1$ to $T_G$ and subsequently transform the intermediate prediction $\hat{I}_G$ with relative transformation ${}^G T_2$. Similarly, we transform source view $I_2$ to camera pose $T_1$. To additionally regularize the

network output, we add supervision by transforming the source views to their respective camera pose: $\bar{I}_2 = f_\theta(I_1, {}^1T_2)$ and $\bar{I}_1 = f_\theta(I_2, {}^2T_1)$. We use a combination of $L_1$ and perceptual losses [18] to get our transformation loss

$$\mathcal{L}_{\text{trafo}}(f_\theta, I_{1,2}, T_{1,2}, T_G) = \sum_{i \in \{1,2\}} ||I_i - \hat{I}_i||_1 + ||I_i - \bar{I}_i||_1 + ||I_i - \hat{I}_i||_{\text{vgg}} + ||I_i - \bar{I}_i||_{\text{vgg}}. \quad (6)$$

**3D feature consistency loss.** We assume the object to be in a canonical view frame which allows us to enforce 3D feature consistency among transformation chains. We define $0°$ for azimuth and elevation as our canonical camera frame. To enable comparison across transformation chain, 3D points $\mathbf{x}$ are sampled at the beginning of each iteration. Given the sampled 3D feature spaces ${}^j\mathcal{V}_i$ of all transformation chains, we enforce feature space consistency with

$$\mathcal{L}_{\text{3d}}(f_\theta, I_{1,2}, T_{1,2}, T_G) = \sum_{i \neq j} ||\mathcal{V}_j - \mathcal{V}_i||_1 \quad (7)$$

for $i, j \in \{1, 2, G\}$.

**Other losses.** To encourage the network to generate spatially meaningful features in 3D space, we also use our object representation network $\Phi$ to predict occupancy for the sampled 3D points. We supervise occupancy prediction using binary cross-entropy between the source masks (we segment the input image instead of using ground truth masks) and the predicted masks. Finally, to encourage the generator to synthesize realistic images, we use a GAN loss $\mathcal{L}_{\text{GAN}}$ with a patch discriminator [16]. Our ful objective function is:

$$\mathcal{L} = \lambda_{\text{trafo}}\mathcal{L}_{\text{trafo}} + \lambda_{\text{3d}}\mathcal{L}_{\text{3d}} + \lambda_{\text{BCE}}\mathcal{L}_{\text{BCE}} + \lambda_{\text{GAN}}\mathcal{L}_{\text{GAN}} \quad (8)$$

where the $\lambda$ parameters are the respective weight of the loss terms.

## 4 Experiments

We evaluate CORN on the task of novel view synthesis for several object classes and on potential applications, namely single-view 3D reconstruction and out of distribution view synthesis on real data. For additional results, we refer the reader to the supplementary document.

**Implementation details.** Hyper-parameters, full network architectures for CORNs, and all baseline descriptions can be found in the supplementary material. Training of the presented models takes on the order of 2 days. Links to code and datasets are available on our project website.

### 4.1 Datasets

**ShapeNet v2.** For novel view synthesis, we follow established evaluation protocols [35, 49] and evaluate on the car and chair classes of ShapeNet v2.0 [4]. The rendered dataset contains 108 images at a resolution of $128 \times 128$ pixels per object, with camera poses sampled from a viewing hemisphere with equally spaced azimuth (ranging between $0° - 360°$) and elevation ($0° - 20°$) angles in 10-degree intervals. Of the 108 rendered images for each object, we select only two images

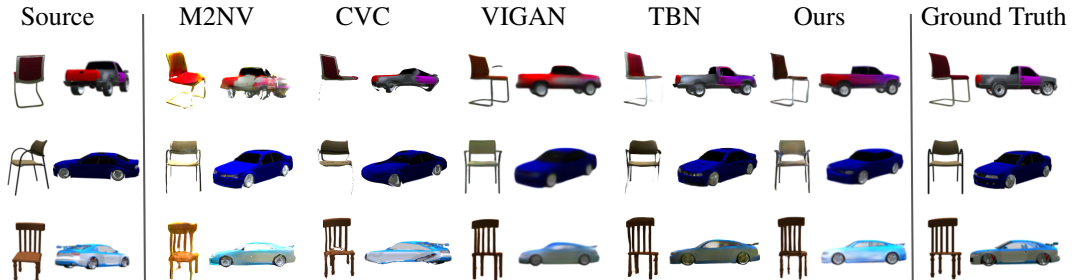

Figure 3: **Qualitative novel view synthesis results.** Our method generates detailed novel views, performing competitively to the baselines. Appearance flow methods fail to generate convincing images for large viewpoint transformations (e.g. top row of M2NV and CVC).

at random per object for CORN training. We evaluate the performance of novel view synthesis on 20,000 randomly generated test pairs of objects in the held-out test dataset. We compare CORN to four baseline methods from recent literature based on three evaluation metrics: 1) $L_1$ distance, 2) structural similarity (SSIM) between the predicted and ground truth images, and 3) learned perceptual image patch similarity (LPIPS) [64], a metric that better replicates human judgment of visual quality.

**Basel Face Model.** We show qualitative results on the Basel Face Model [11] with face models showing a range of expressions. We generate 33 renderings each from 10,000 face models with camera poses sampled from a hemisphere in front of the face with azimuth angles ranging between ($\pm 50°$) and elevation ($\pm 20°$). Appearance and expression parameters are sampled from a zero-mean Gaussian distribution with a standard deviation of 0.7. We use constant ambient illumination and a variation of expression and appearance parameters. After training, we evaluate our model on a holdout test set. CORNs capture the appearance and orientation of the face model and expression parameters, as seen in Fig. 4. As expected, due to the deterministic implementation, CORNs reconstructs objects resembling the mean of all feasible objects, decreasing performance for instances showing considerable divergence from the mean shape. For additional results, see the supplementary material and the video.

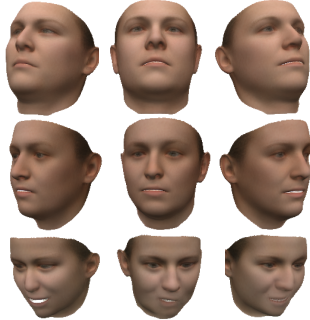

Figure 4: **Qualitative results on the Basel Face dataset.**

## 4.2 Results

**Novel view synthesis**. We evaluate our network on the task of transforming a single source-view into a target camera view. Table 1 and Fig. 3 show quantitative and qualitative results respectively. Despite using no 3D or 2D target view supervision and only two source images for training, our method performs competitively (up to within 2% of the best score) or even outperforms other methods on the LPIPS score against the supervised approaches, demonstrating CORN ability to generate meaningful object representations from a fraction of the data. The appearance flow-based methods fail if the viewpoint transformations are large (top row). Our model qualitatively preserves fine details and generates meaningful results for missing parts of the object. Using the two source images for direct supervision (CORN w 2VS) without learning a consistent 3D representation and using transformation chains decreases model performance. Similarly, relying solely on a global object description vector (CORN global) decreases performance. For additional ablations of individual loss function terms, please see the supplementary material.

Table 1: **Quantitative novel view synthesis results.** We report mean and standard deviation of the $L_1$ loss (lower is better), structural similarity (SSIM) index (higher is better) and learned perceptual image patch similarity (LPIPS) (lower is better) for several methods. Our model achieves competitive results, using $50\times$ less data and only two input images per object for self-supervision.

| Methods | Car | | | Chair | | |
|---|---|---|---|---|---|---|
| | $L_1(\downarrow)$ | SSIM ($\uparrow$) | LPIPS ($\downarrow$) | $L_1(\downarrow)$ | SSIM ($\uparrow$) | LPIPS ($\downarrow$) |
| M2NV [49] | 0.139 | 0.751 | 0.238 | 0.114 | 0.738 | 0.217 |
| CVC [5] | 0.091 | 0.802 | 0.149 | 0.124 | 0.741 | 0.207 |
| VIGAN [59] | **0.0524** | **0.860** | 0.130 | 0.121 | 0.746 | 0.161 |
| TBN [33] | 0.0578 | 0.856 | 0.095 | **0.092** | **0.792** | **0.138** |
| CORN w 2VS | 0.073 | 0.819 | 0.120 | 0.157 | 0.689 | 0.245 |
| CORN global | 0.069 | 0.827 | 0.109 | 0.157 | 0.703 | 0.216 |
| CORN | 0.063 | 0.838 | **0.094** | 0.132 | 0.722 | 0.180 |

**Continuous scene representation.** We compare our method against SRN [47], a state-of-the-art continuous scene representation network. To allow a fair comparison, we use only objects at the intersection of the two evaluation protocols. We use a single source image for both models to

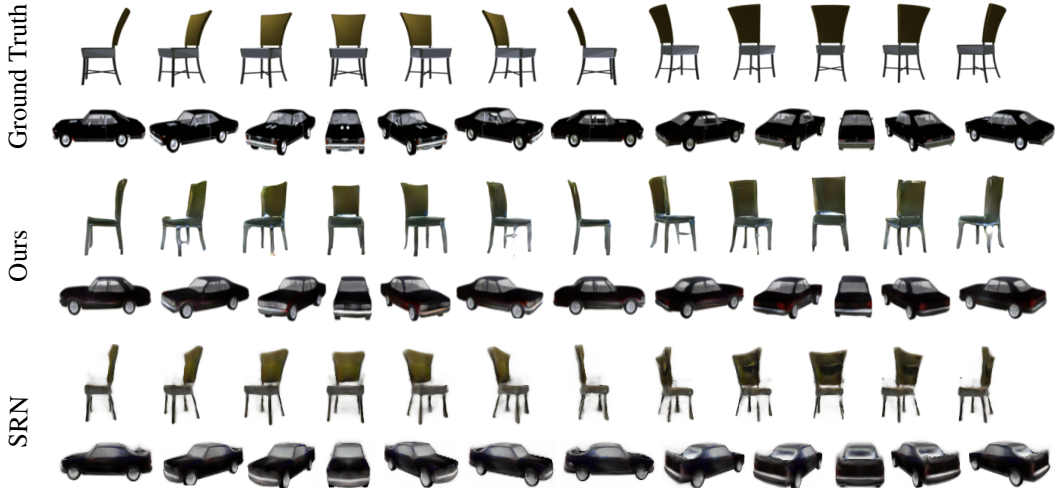

Figure 5: **Quantitative results of view sequence generation.** We used a single image to predict 108 target views. Our model synthesizes images with a higher level of detail, models transient object properties, and shows better form fit.

reconstruct the scene representation before generating 108 views (ranging between azimuth $0° - 360°$ and elevation $0° - 20°$ angles).

Table 2 and Fig. 5 show the qualitative and quantitative results respectively. Our method outperforms SRN, even without target view supervision, using only two images per object, compared to 50 for SRN. Our model does not require latent code optimization during inference, which dramatically reduces the inference time from about 5 minutes (SRN) to milliseconds per image.

Table 2: **Continuous representation results.** We evaluate the ability of SRN [47] and CORN to synthesize 108 novel views from a single input image on car and chair objects. Our method outperforms SRN on both datasets.

| Methods | Car | | | Chair | | |
|---|---|---|---|---|---|---|
| | $L_1(\downarrow)$ | SSIM ($\uparrow$) | LPIPS ($\downarrow$) | $L_1(\downarrow)$ | SSIM ($\uparrow$) | LPIPS ($\downarrow$) |
| SRN [47] | 0.090 | 0.797 | 0.144 | 0.160 | 0.708 | 0.232 |
| CORN | **0.067** | **0.831** | **0.103** | **0.1333** | **0.725** | **0.181** |

### 4.3 Applications

**Out of domain view synthesis.** Results reported so far are on synthetic datasets where the input images are rendered from 3D CAD models. To test the generalization performance of CORNs to real data, we evaluate our model trained with the ShapeNet car objects on the Cars [19] dataset. This dataset contains a wide variety of real car images taken from natural scenes. Note that we did not retrain our model on this dataset. Fig. 6 shows the novel view synthesis of objects given real input images as input. Our method preserves local geometric and photometric features in this challenging setup. This experiment suggests that our model can synthesize images from different dataset distributions, indicating some domain transfer capability.

**Single-view 3D reconstruction.** In addition to novel view synthesis, our method's possible applications include single-image 3D reconstruction. We synthesize $N$ novel views on the viewing hemisphere from a single image. From these images, we sample $k$ 3D points uniformly at random from a cubic volume in a similar procedure to the one described in Section 3.2. Our goal is to predict the occupancy of each of these $k$ points. To accomplish this, we project each point onto the synthesized images and label it as occupied if it projects to the foreground mask. Fig. 7 shows the 3D reconstruction results from the input images. Our method captures the overall structure of the

Input                          Novel Views

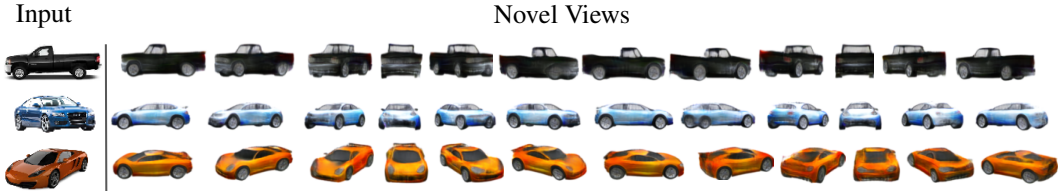

Figure 6: **Qualitative results of novel view synthesis on real data.** Our model generates high quality views of previously unseen data. We use the model trained on ShapeNet and evaluate on the Cars [19] without retraining.

Input       Ours       Ground Truth       Input       Ours       Ground Truth

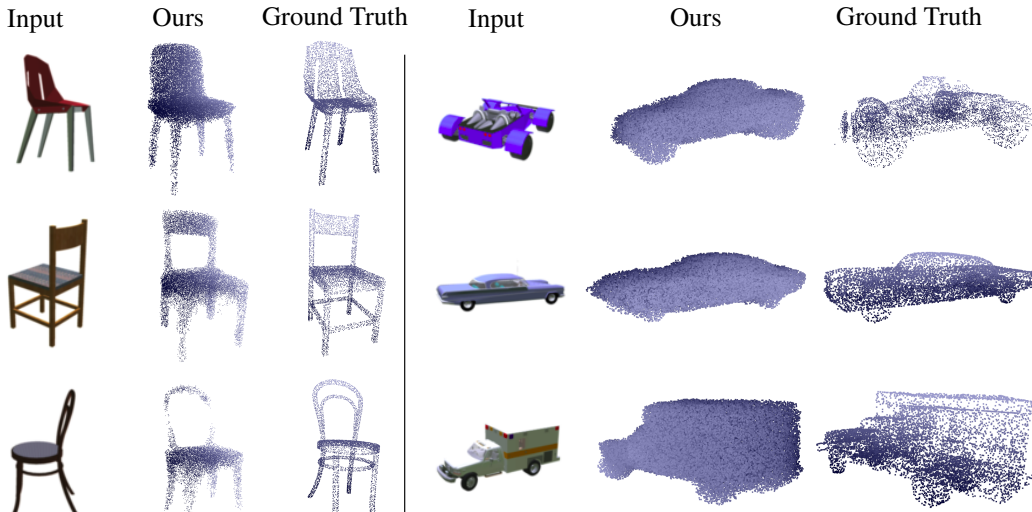

Figure 7: **Qualitative single view 3D reconstruction results.** The synthesized views can be used to produce high quality 3D reconstructions from a single input image.

objects, and in most cases, their fine-level details. In our experiments we synthesize $N = 15$ images and sample $k = 10^5$ points in 3D space.

## 5   Discussion

We introduced CORNs, a continuous neural representation for novel view synthesis, learned without any 3D object information or 2D target view supervision. Our system's key component is the use of transformation chains and 3D feature consistency to self-supervise the network. The resulting continuous representation network maps local and global features, extracted from a single input image to a spatial feature representation of the scene. Incorporating a differentiable neural renderer enables the synthesis of new images from arbitrary views in an end-to-end trainable fashion. CORN requires only two source images per object during training and achieves comparable results or outperforms state-of-the-art supervised models with $50\times$ fewer data and without target view supervision. We demonstrate our model's applications for novel view synthesis, single image 3D object reconstruction, and out of distribution view synthesis on real images.

There are several exciting possible avenues for future work. Using two input views during training to regularize the 3D representation by imposing consistency across the input training images is critical to our method's success. We intend to investigate whether such regularization can be achieved with only one training image per object. However, the naive solution of using CORN with only a single image fails, as cyclic consistency collapses the model to the trivial solution (the identity mapping). Currently, CORNs achieve high confidence predictions from two randomly chosen training images. Sampling views more intelligently, i.e., across aspect graph [8] event boundaries could improve performance. Finally, CORN operate on synthetic data, without natural backgrounds, and with defined camera poses. Future work may alternatively integrate CORN with pose estimation and background modeling models.

## Broader Impact

Our approach to learning novel view synthesis has the immediate possibility of enabling augmented and virtual reality applications [37, 20]. The limited requirement of only two images per object will make these techniques applicable beyond synthetic data. As with any generative model that provides tools for image manipulation, we too run the risk of producing fake visual content that can be exploited for malicious causes. While visual object rotation itself does not have direct negative consequences, the mere fact that such manipulations are possible can erode the public's trust in published images [34]. Image manipulation is not a new phenomenon, however, and there has been research trying to detect manipulated images [41, 2] automatically. Still, more work on and broader adoption of such techniques is needed to mitigate image manipulation's adverse effects.

## Acknowledgments and Disclosure of Funding

Funding provided in direct support of this work came from the UMII MnDrive Graduate Assistantship Award a LCCMR grant and NSF grant 1617718. The authors acknowledge the Minnesota Supercomputing Institute (MSI) at the University of Minnesota for providing resources that contributed to the research results reported within this paper. URL: `http://www.msi.umn.edu`

## Footnotes

[1]Project page: nicolaihaeni.github.io/corn/

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
