[Supplementary Material]

# Continuous Object Representation Networks: Novel View Synthesis without Target View Supervision

**Nicolai Häni**       **Selim Engin**       **Jun-Jee Chao**       **Volkan Isler**
{haeni001, engin003, chao0107, isler}@umn.edu
University of Minnesota

## Contents

## 1 Additional Results

This section presents additional qualitative results of our models for each evaluation scenario presented in the paper.

### 1.1 Loss function ablations

We use the ShapeNet v2. [1] chair category and train models by ablating parts of the loss function. Due to the long training times per model, we provide ablations only on the chair category. Our transformation chain and 3D feature supervision are shown to be necessary, as they improve the baseline's performance (Tab. 1). Similarly, performance degrades if the GAN loss or the perceptual loss functions are omitted.

Table 1: **Loss function ablation.** We report mean and standard deviation of the $L_1$ loss (lower is better), structural similarity (SSIM) index (higher is better) and learned perceptual image patch similarity (LPIPS) (lower is better). The ablation shows the importance of our proposed loss function.

| Methods | Chair | | |
|---|---|---|---|
| | $L_1(\downarrow)$ | SSIM ($\uparrow$) | LPIPS ($\downarrow$) |
| CORN w.o $\mathcal{L}_{\text{trafo}}$ | 0.211 | 0.656 | 0.341 |
| CORN w.o $\mathcal{L}_{\text{GAN/VGG}}$ | 0.141 | 0.714 | 0.216 |
| CORN w.o $\mathcal{L}_{\text{GAN}}$ | 0.139 | 0.715 | 0.208 |
| CORN w.o $\mathcal{L}_{\text{3D}}$ | 0.133 | 0.724 | 0.182 |
| CORN | **0.132** | **0.722** | **0.180** |

## 1.2  Additional view synthesis results

We show additional qualitative results of our model on the ShapeNet v2 [1] cars and chairs categories on the task of transforming a single-source view to a target camera pose.

Figure 1: **Additional qualitative results.** Given a single source image and a target camera pose, our method generates fine level details even for unobserved parts of the object. We provide the ground truth view of the objects for reference.

In Fig. 2, we show additional results on the task of creating a sequence of consecutive target views from a single input image.

Figure 2: **Additional rotation results.** We used a single image to predict 108 views. Compared to the ground truth image sequence (top row per model) our model (bottom row per model) generates target views with high detail.

## 1.3 Additional results for out-of-domain view synthesis

In Fig. 3, we show additional results on using our pretrained model on out-of-domain images. We take the model pretrained on the ShapeNet v2 dataset and apply it on images from the Cars [4] dataset without retraining. We observe that the model predicts the structure and appearance correctly; however, the detail degrades compared to the synthetic ShapeNet images. Further, the generated images inhibit the training set's characteristics, with the model being unable to model specularity or all the adequate levels of details.

Input                                 Predictions

Figure 3: **Additional synthetic → real results.** Given a single image of a real car, we predict 108 novel views. We use our model trained on the synthetic ShapeNet dataset without retraining.

## 1.4  Additional results on the Basel Face dataset

In Fig. 4, we show additional results on the task of creating a sequence of consecutive target views from a single input image on the Basel Face dataset [3].

Figure 4: **Additional face rotation results.** We used a single image to predict 33 views.

## 1.5 Additional single view 3D reconstruction results

In Fig. 5 we show additional examples of using synthesized views for single image 3D reconstruction. We use a single image to predict 15 novel views. We re-project 3D points to each foreground mask to receive occupancy probabilities.

| Input | UniCORN | Input | UniCORN |
|-------|---------|-------|---------|

Figure 5: **Additional 3D reconstruction results.** Given a single input image our method can synthesize images from multiple viewpoints to generate the 3D reconstruction of an object.

## 2 Reproducibility

Here we give more information about the precise architecture used to build our model.

### 2.1 Network Architecture

**Spatial and global object feature network.** We use a pretrained ResNet-18 for global and upsample blocks for the local feature extraction. In particular, we use the setup in Fig. 6a.

**Continuous function network.** We use an MLP for the continuous function network. In particular, we use the setup in Fig. 6b.

**Neural renderer.** Our neural renderer based on [9] uses disks of radius 2 pixels for splatting and stores 16 points per pixel for z-buffering. Please refer to [9] for additional details.

**Refinement network.** We use a UNet for the refinement network, containing four down/upsample blocks with skip connections. In particular, we use the setup in Fig. 6c.

### 2.2 Implementation details

The models are trained with the Adam optimizer, learning rate of $0.0002$, and momentum parameters $(0, 0.99)$. Empirically, we found $\lambda_{rec} = 10, \lambda_{3d} = 1, \lambda_{bce} = 1$, and $\lambda_{gan} = 0.1$ to lead to good convergence. The models are trained for $45$ epochs (chair) and $5$ epochs (car). We implemented our models in PyTorch [6]; they take 1-2 days to train on 4 Tesla V100 GPUs. For architectural details please see the supplementary material. Code, data, and model files can be accessed on our project page.

Figure 6: **CORN architecture:** **(a)** Our feature extractor uses a pretrained ResNet-18 for global feature extraction (left). For local feature extraction (right) we use four upsampling blocks with skip connections (red arrows). **(b)** Our continuous function network takes as input a position encoding, local features and global features and produces a 3D feature at the given spatial location and an occupancy probability. **(c)** Our refinement network contains a UNet with four down/upsampling blocks.

(a) Feature extractor

(b) Continuous function network

(c) Refinement network

## 3 Baselines

On the task of novel view synthesis, we compare CORN against five current methods that use 2D target view supervision. In our first experiment, we compare our approach against [8, 2, 10, 5], which share a common evaluation protocol. At inference time, the models are presented with a single source image and generate a fixed target view. To compare our method against SRN [7], we use a single image to generate the 3D object representation. From this representation, we synthesize all 108 views and compare them against the ground truth. Note that all these baseline methods have access to the ground truth target images, while our method does not.

### 3.1  M2NV:

Multi-view to novel view (M2NV) [8] predicts the flow of multiple source images to a single target camera pose. During training, the network predicts each flow map's confidence scores and aggregates the pixel values in the final target view. At test time, the method can take an arbitrary number of input images.

**Training.** We use pretrained model weights for the ShapeNet chair and car categories, made available by the authors.

**Testing.** For novel view synthesis on the test set, the model receives a single input view and generates the requested target view.

### 3.2 CVC:

Monocular neural image-based rendering with continuous view control (CVC) [2] is another appearance flow method. The approach estimates a 3D latent space that is implicitly rotated to the target camera pose. A decoder uses the rotated latent space to output depth predictions. Using perspective projection, they attain dense per-pixel correspondences, which are used to warp the pixels from the source to the target view.

**Training.** We use pretrained model weights for the ShapeNet chair and car categories, made available by the authors.

**Testing.** For novel view synthesis on the test set, the model receives a single input view, and we generate the requested target view. In [2], this method is evaluated for target views that are within $[-40°, 40°]$ rotation from the source image. We evaluate it using the provided models over the entire range of the dataset.

### 3.3 VIGAN:

View independent generative adversarial networks (VIGAN) [10] predicts novel views by learning a view independent feature map. The network generates novel views in the target camera frame by combining this feature map with the target camera pose. A decoder network synthesizes the final image.

**Training.** We reimplemented VIGAN and modified the loss functions due to the non-convergence of the original model. We remove all loss functions, except reconstruction, cyclic consistency, pixel losses. The model was trained from scratch on our training dataset for 100,000 iterations (chairs) and 1,000,000 iterations (cars) with the hyperparameters proposed in the original paper.

**Testing.** For novel view synthesis on the test set, the model receives a single input view, and we generate the requested target view.

### 3.4 TBN:

Transformable bottleneck networks (TBN) [5] learn a discrete voxel representation from multiple 2D images. The model disentangles viewpoint transformations and the objects' appearance by introducing a transformable bottleneck layer. This bottleneck layer is transformed with arbitrary SO(3) rotations and reprojected to the target camera pose to guarantee multi-view consistency.

**Training.** We use pretrained model weights for the ShapeNet chair and car categories, made available by the authors. This method is trained using four input images.

**Testing.** For novel view synthesis on the test set, the model receives a single input view, and we generate the requested target view.

### 3.5 SRN:

Scene representation networks (SRN) [7] are closely related to our method, learning a continuous scene representation with multi-view consistency.

**Training.** We use existing models, trained on 50 source images per object.

**Testing.** We optimize for the latent vectors of objects in our test set until convergence using the provided code and hyperparameter configurations.