[Reviews · NeurIPS 2020]

Review 1

Summary and Contributions: UPDATE: I've read the rebuttal, and I appreciate the results from the supervised method and the preliminary results on the more realistic dataset. I am happy to increase my score to 7. ### This paper proposes a new approach for learning 3D object representation, in this case in a continuous form, where the representation consists of a global and a local input representation and a query function that extracts feature vectors at arbitrary 3D points. Additionally, the paper introduces a specific architecture for the task and a novel optimization objective based on cycle consistency. The model is evaluated only on a single dataset (ShapeNet), by doing standard novel-view synthesis, but also on 3D reconstruction from single image (also ShapeNet) and on out-of-domain generalization, where a model trained on synthetic data (ShapeNet again) is evaluated on real data (real cars). The authors claim that their method is unsupervised (which I do not believe is the case, more on this in "correctness") and use 50x less data than baselines (again, this is not true, see "correcntess") and compare with supervised method such as VIGAN, TVN and CVS and also SRN, achieving comparable results. To sum up, while the method is very interesting and performs well, it contains incorrect claim, which forces me to recommend rejection in its current state.

Strengths: - The method as a whole achieves decent results, seems interesting and easy to built upon, and is quite relevant to the community as evidenced by a lot of cited work in the area. - Empirical evaluation for 3D reconstruction is a clever and well-executed idea, and together with out-of-distribution generalization does showcase the strengths of this method compared to the baselines. - I think the method is novel: while I am not an expert in the area, I do think that the particular method of lifting global and local features into continuous scene representation has not been explored before, and certainly not in conjunction with such reprojection-error-based losses. - The method cleverly uses a pre-trained segmentation network (no idea which one, though) to segment-out objects and formulate an additional "occupancy" loss in the 3D space, which improves results.

Weaknesses: - The method is evaluated only on a single dataset, and therefore it is difficult to say when and how it breaks as compared to other methods. - The chosen architecture is quite complicated (local features from a ResNet, global ones from a UNet, differentiable renderer, UNet for a refinement network), and there are no ablation studies that would examine the significance of different components. - The authors claim that the method is a) unsupervised and b) that it uses 50x less training data than baselines. Both are incorrect, and here is why. a) The method is trained by using two input images taken from known camera poses. It is "unsupervised", because it does not use a "target" image and an associated camera pose to compute its loss, which is in contrast to baselines, which do. However, given that this method uses two images with poses, it might as well use a supervised loss where representation is extracted from a single image, and the second image is used as the target one. This would be enough to train GQN, say. In fact, the cycle-consistency losses require bi-directional transformation between the two images, which requires the known poses, and is very similar to a directly-supervised loss. -- Additionally, using a pre-trained segmentation network, while being clever, is like cheating in this case. Such networks are trained with supervised data, and hence it is incorrect to claim that a method based on a segmentation network is unsupervised. b) The 50x less assumption is based on 108 images per object in the dataset, while this method only uses 2 images per object. However, the method still uses all 108 images in the training set, and it is trivial to show that with stochastic minibatch training this is the same as taking a monte-carlo approximation of using all 108 images at once. The authors do not show what happens when they actually train on 50x less data, which makes this claim incorrect.

Correctness: See last point in "weaknesses", since there was not enough space to address everything here. - l57-58 say that "implicit 3D representations do not allow extraction of the scene's 3D structure". This is not true in e.g. NERF (as cited by this work), where representation is implicit but the rendering is explicit, and therefore it is possible to extract full 3D structure. Please amend this sentence. - l68-69: saying that continuous representations represent a scene as a binary classifier or SDF is too narrow and therefore not correct as shown by NERF, which is neither of those and yet is continuous. Please amend this sentence. - l97-98 say that global representation z captures object's semantics while local representation captures details. Yet, there is no investigation into this aspects in the paper. Please either provide a (n empirical) proof or remove this claim.

Clarity: The paper is clearly written. Minor remarks: - l96: what is an "assymetric" FCN? - it's only at l137-138 that we get to know how many and from what distribution the 3D points are sampled from, despite them being mentioned in at least two locations (l107 and caption of fig2). Could you please either explain it fully or not mention at all earlier?

Relation to Prior Work: The paper discussed relation to prior art in novel view synthesis, 3D scene representation and generative modelling, which is very well done. My only suggestion is that l80-83 should be moved under the NVS section.

Reproducibility: No

Additional Feedback: Questions: 1. How would you apply your method to dense images with e.g. backgrounds?\ 2. Both local features and 3D features layer on use 64-dim vectors as representation, which seems to be very high given the resolution of images and of the possible 3d representation. Why so many? Does setting this number lower influence results much? 3. l113-114 mentions that local features for a specific location are extracted using bilinear sampling. You could also use attention as in e.g. Kim et. al. "Attentive Neural Processes". Would that be better/worse, and why? 4. Instead of relying on a refinement network to fill-in the blanks, you could compute where points should be sampled to avoid blanks. Would that help, perhaps with and without removing the refinement network? Other remarks: - I'd rename sec 3.3 as "Fourier point encoding" to be more clear.


Review 2

Summary and Contributions: This paper proposes a method for novel view synthesis that only requires two input images of an object during training. The paper claims that the approach does not require 2D target view supervision, even though it still uses the two input views in the cycle consistency term and therefore is using the 2D pixel supervision. The approach uses continuous representation network to encode the input image into a canonical 3D representation space, from which a target view is rendered. This is generated target view is then used to generate the two input views, forming a cycle consistency loss. The paper also uses mask re-projection loss and other GAN losses. The paper uses a combination of implicit 3D representation and a point-cloud based neural renderer. The results are experimented on standard benchmarks where the proposed approach does obtain qualitatively nice results.

Strengths: - Combining implicit neural representation for 3D with point cloud rendering is a nice way to harvest the strength of both reprsentation. In particular this approach allows much faster rendering than compared to method like SRNs. I think this is the most interesting contribution in this paper. - The approach is only trained on two views of a training instance, but the results look qualitatively better than SRN, a recent SoTA approach that uses 50 images of an training instance (although this experimental setup is rather toy, since it's a synthetic dataset, why artificially add constraints? If the point is to show that this can be trained on less data, then it should be trained on real images.) - Evaluated on real, albeit a very toy / clean images of real cars. - Good broader impact statement

Weaknesses: The claim made in the paper is arguable and unclear. The paper is written with key details missing and misleading wording. - "No 2D or 3D supervision". First, most view synthesis work such as SRN [46], CVS [5], Zhou et al [63] and many others do not use 3D supervision. Often that's implied via view-synthesis method. Therefore adding unsupervised in the title is also misleading and unecessary (see point below that if you require at least 2 views no reasxon not to use the direct view synthesis loss). Second, I think the characterization that this approach does not need 2D supervision is misleading and just not true. It uses the reconstruction to the two input views through the cycle consistency loss. I think the authors mean this in that there is not (source, target) supervision. However this is also not true, this the view-synthesis loss from the two input view is used (line 156) and when removed in ablation hurts L1 performance the most. If a method truly did not require any view supervision (here which comes from source1, source2), this means the approach can be trained from single-view instances alone, and the approach presented cannot handle this case. Furthermore, I do not see the motivation to remove the 1->2 view synthesis loss (which this paper actually uses). If you have two views of the object, why not use that to supervise. The motivation seems superficial. - Writing is missing the key detail of how is the target camera pose $T_G$ being specified, which is supposed to be the canonical view. This is a big ommision. These are quite serious issues that the paper ought to improve upon for any publication. Here are other issues: - SSIM is really not an informative metric. It's not adding much to the evaluation at all. The values are very close for most experiments for all methods. L1 is also better but not the best, please include LPIPS in the evaluation metric. - Figure 2 contains the Mapping equation, but this is not explained in the main document. - Figure 4: Highlight which image is the input image.

Correctness: No. please see above.

Clarity: No. please see above.

Relation to Prior Work: Not really. It's not very well discussed because all discussion is in relation to this lack of 2D/3D supervision. The paper says that the The paper should really emphasize the benefits of combining implicit 3D representation with pointcloud based renderer, which sets this paper apart from previous work.

Reproducibility: No

Additional Feedback: The results do look nice and it is impressive that it only uses 2 views of an instance during training to obtain this result. However the claims made in this paper are not accurate of what actually happens in the paper. There are strength here (low sample, fast rendering) but the paper requires a whole new story and a re-write. ========== Update My initial rating was a bit harsh due to the misleading claim also raised by other reviewers. This is clarified, and I appreciate that the authors incorporated LPIPS and it seems to make a difference in the response (Table 1 in response is very good). As I mentioned in the original review, I think there are advantages to this paper such as fast rendering, good results with less supervision. However, after discussion with other reviewers, I'm still at borderline: - I am still not really happy with the claim of not using any 2D supervision, the author's answer is that "unsupervised" refers to "no target view", but this work uses direct view synthesis loss which is to predict one source view from the other view, making that a known target view. This is just be poor wording, no intermediate pose would be more accurate.. if we are going to accept this paper we should emphasize that "unsupervised" should be removed. - Real experiment that are illustrated in the response is very promising, however details are missing and while promising does not change my opinion on the paper. If a similar result to Table 1 is available for real images this would be a very strong result. Therefore I am updating my rating to 6: Marginally above ================


Review 3

Summary and Contributions: This paper presents an algorithm to predict a continuous 3D representation of an object for novel view synthesis from a single input image by only using two observed images of each object during training, as opposed to prior work which used many more images for supervision during training. The paper's main contribution is using additional losses to encourage the predicted features to be consistent at the same 3D location as well as cycle-consistency losses. Comments after rebuttal/discussion: I agree with other reviewers that the paper's claims that the method is "unsupervised" is misleading, and requires rewriting. Regarding the rebuttal, I like how Table 1 shows that using the cycle consistency losses seem to improve results over just using the direct view supervision losses. I don't think the rebuttal really addressed my concerns regarding the paper's specific choices of 3D representation and differentiable rendering functions, and I think that this is a substantial issue.

Strengths: I think that the cycle consistency loss is a good idea that can also be incorporated in other settings that predict 3D representations for view synthesis. The qualitative and quantitative results seem on-par with other recent algorithms.

Weaknesses: Many of the design choices seem specialized to simple shapenet-style objects (including the assumption that all scenes have a canonical orientation and the use of masks for occupancy supervision, and it is difficult to imagine scaling this approach up to more realistic scenes (such as shown in SRN and NeRF). I think that renderer/3D representation could benefit from more precise description. Section 3.2 describes the representation as a continuous field of occupancy probabilities and 64-dimensional features. However, in section 3.4, the rendering function is described as using splatting, soft rasterization, and alpha compositing. This is a bit confusing to readers, since an occupancy field can be rendered as a surface by finding the surface crossing for each pixel, but this method instead would softly project all 3D points onto the image plane without correct occlusions. I suggest that the authors discuss the choices of 3D representation and rendering more thoroughly. Regarding the use of soft rasterization for differentiable rendering: why use a biased/approximate rendering function instead of the differentiable occupancy field rendering functions in https://autonomousvision.github.io/differentiable-volumetric-rendering/ [CVPR 2020]? I suggest that the authors provide additional discussion on the refinement network. It is not quite obvious to me why the 3D representation would not contain information about parts occluded in the input images. Shouldn't the model learn to predict meaningful occupancy probabilities and features vectors for these occluded regions? For example, the PIFU work which uses a similar representation doesn't need a refinement network. The refinement network can cause the representation to not be multiview consistent, as seen in the warping appearance of objects in the results video. Supervising the occupancy using binary cross-entropy between the predictions and masks for the input view sounds unintuitive. When views are close together, this would mean that a large portion of 3D space (the "silhouette hull" way outside of the actual 3D footprint of the object) would be supervised to have occupancy. Could the authors please discuss this choice more? It would also be interesting to ablate the refinement network, and whether the representation itself encodes occluded content or if it is just reconstructed within the refinement network.

Correctness: Yes, I believe that the paper, algorithm, and experiments are technically correct.

Clarity: Yes, the paper is overall well-written and easy to understand.

Relation to Prior Work: The paper mostly does a reasonable job discussing prior work, but I have a few suggestions. The proposed 3D representation (features at each pixel which are concatenated to the depth and passed into an MLP that predicts a continuous representation and occupation probability) seems quite similar to PIFU [41], and this should be discussed more.

Reproducibility: Yes

Additional Feedback: line 21: induces --> introduces line 53: this wording may be confusing to readers since "implicit" is also frequently used to describe implicit surfaces (surfaces defined as the level set of a scalar function) line 66: vectors --> vector line 67: may also be worth discussing [27] (NeRF) here, since they use a different continuous scene representation than those discussed in this sentence line 134: remove extra period Bibliography edits: [55] appeared at CVPR 2020, [27] will appear at ECCV 2020, some references contain conference abbreviation (CVPR) while others don't.


Review 4

Summary and Contributions: ----------------------------- # Update: Thank the authors for the clarification in the response. Some of my concerns have been addressed. But I still think that since the main point of this paper is to use less supervision, it would be great to provide insight by designing experiments w.r.t. supervision or directly showing more results in complex real objects. In the author response, the single-image case is shown, but it is still not clear whether the performance really saturates at 2 images. If it does not saturate, this means it is the implicit representation method itself as proposed in NERF being good, but not the proposed method being effective in reducing supervision. After all, the visual performance degrades so much compared to original NERF, even shown in the teaser figure (in the last case the input-output cars are in close angles but high-freq details are simply gone). ------------------ This paper proposes a method for novel view synthesis using only 2 images without access to 3D groundtruth in training time.

Strengths: Introducing implicit representation as latent 3D representation indeed has the promise to solve scalability problem that explicit representations have to face. Enabling learning implicit representation from few images is important and interesting.

Weaknesses: Some important evaluations are missing. Regarding the main contribution (few images used in training), more evaluation on supervision should be present. By design, only one image is actually enough to obtain cyclic loss. Since the model is evaluated on shapeNet, a single image per instance has been shown enough to learn shape priors for a single object category. On the other hand, what if more images are available? Does the performance saturate at two images (which is good), or it is at the cost of quality? Comparison to the state of the art is incomplete. SynSin [55] is very relevant (the setting of [55] is also very closed to this paper) to this work while it is not compared with. In fact, I think the difference between SynSin and this work is mainly in the latent 3D representation (SynSin used point cloud), but its visual quality is more compelling and it generalizes to complex real indoor scenes. In contrast, the visual quality of the rendered results in this paper is not as compelling as expected. For example, the high frequency details are missing, see the last row of Figure 1(b) where the painting on the car is gone. This is somehow unexpected and confusing, as positional encoding is added to further enable the model to capture high frequency.

Correctness: Some important technical details are not clear. - L107, "We sample 3D points x uniformly at random". It is not clear how the points are sampled, randomly or uniformly in grids? - L118, "we also predict occupancy prob .. as binary masks". How is it used? To constrain the sampling of 3d points? - L140 "using a neural renderer based on [55]". How is the neural renderer like? If it is exactly the same as [55], what is the hyperparameters values?

Clarity: Basically yes, but there are some technical issues as stated above.

Relation to Prior Work: Basically yes, although I believe more discussion about connection to SynSin should be included, as the technical pipeline is actually quite akin to it.

Reproducibility: No

Additional Feedback: How is it generalized to more complex scenes, e.g. indoor scenes?

[Author Response · NeurIPS 2020]

We thank the reviewers for their time and insightful comments. We were able to address most of them, which helped
improve the paper. We focus on the major comments below, and minor ones will be addressed directly in the paper.

**Direct supervision between the two source images [R1, 2, 4].** This is a viable approach; however, it does not
generalize as well as our network, as our method reconstructs the source images from unseen target viewpoints, ensuring
visual coherence for these views. To support this claim, we designed the proposed approach and retrained the network.
Our approach outperforms direct supervision by over 20% (Table/Fig. 1), confirming our approach's superiority.

| Methods | Car | | | Chair | | |
|---|---|---|---|---|---|---|
| | $L_1(\downarrow)$ | SSIM ($\uparrow$) | LPIPS ($\downarrow$) | $L_1(\downarrow)$ | SSIM ($\uparrow$) | LPIPS ($\downarrow$) |
| Supervised | 0.0368 | 0.8481 | 0.1283 | 0.0786 | 0.72 | 0.2501 |
| Ours | **0.0338** | **0.8632** | **0.1029** | **0.0663** | **0.7568** | **0.1766** |

Table 1: Direct supervision vs. Ours

Figure 1: Supervised (left), ours (middle), ground truth (right)

**Claim of using no 2D supervision is misleading [R1, 2].** Unsupervised in the paper refers to no target view supervision.
Our method does not use any supervision for target viewpoints, only self-supervision on the two source views. We have
clarified this point in the paper.

**Perceptual evaluation metric [R2].** This is a great point, and we have added the suggested metric to our comparison
(Table 1). Full results will be added to the paper.

**Model has access to 108 training images during training [R1].** We select two images per object *before* training and
only use these for training (LN 187-188). We rewrote the section to avoid confusion.

**Evaluation on other datasets than ShapeNet [R1, 3].** This is a great suggestion and we show prelimi-
nary results on the real capture dataset (Fig. 2). Additional details and results will be added to the paper.
16

**Comparison to SynSin [R4].** SynSin allows rotations of limited range ($\pm 20°$) since
it uses a depth point cloud for the 3D features. In our method we generate novel views
on the entire viewing hemisphere and therefore did not include this comparison.

**Neural renderer description [R3].** We agree with the reviewer and added additional
details of the rendering function in the supplementary material.

Figure 2: Source (left), pre-
diction (middle), target (right)

**Choice of neural rendering function [R3].** This is a very active research area, without
a clear consensus on which rendering functions are preferable (see the recent summary
report "State of the art in neural rendering"). We chose a rasterization based rendering
function as it is fast, differentiable, and the splatting of features to the image plane allows for good gradient flow.

**Refinement network and adaptive sampling [R3].** The refinement network fills in gaps and improves rendering
quality of the rasterization based neural renderer. Adaptive sampling is an interesting suggestion, and our network
would likely benefit from such a step. We will keep this as future work.

**Target poses $T_G$ specification [R2].** Poses are represented with spherical coordinates represented by azimuth and
elevation. The target pose is selected randomly and given to the network as an input (Fig. 2 in the paper).

**More realistic scenes [R3, 4].** This is a research direction that we would like to investigate in future work (LN 254).
We plan to investigate model generalization to natural images with background and scenes containing multiple objects.

**Bilinear sampling vs. Attentive Neural Processes [R1].** Using "Attentive Neural Processes" looks like a viable
option, but we have not investigated this as an alternative and do not know if this would improve performance.

**Single image cyclic consistency [R4].** This is technically correct. However, the network collapses to a trivial solution
(the identity function) when using a single image. We have added this to the paper.

**Supervising the occupancy using binary cross-entropy [R4].** For a single instance, the visual hull can be outside of
the object's 3D footprint. However, we find that the multi-view consistency enforces the predicted occupancy features
to be coherent with the object's actual 3D structure. We also have preliminary results for extracting the 3D model using
the occupancy features.

**Performance saturation [R4].** We think that using multiple views will improve performance, but we have not
investigated this claim nor the issue of performance saturation. Our approach shows that novel view synthesis is feasible,
even if only two images per object are available.

**Segmentation network [R1].** We do not use any (pre-trained or not) segmentation networks in our method.

[Meta-Review · NeurIPS 2020]

Four qualified referees have carefully reviewed the paper. They agree that the method is interesting and the iew synthesis results on ShapeNet are strong given the small amount of training data used. On the downside, reviewers question "unsupervised" in the name of the method (even though it requires images with ground truth camera poses) and the fact that experiments are limited to relatively simple synthetic data. If sample efficiency is the point, an evaluation on real data would make much sense, as well as experiments focused on demonstrating sample efficiency, e.g. train the proposed method and some baselines on varying amounts of data and show that the proposed method degrades more gracefully. After the rebuttal and discussion, the reviewers agree that the merits of the paper overweight the issues, so I recommend acceptance. However, the authors are strongly encouraged to reconsider using "unsupervised" in the title and encouraged to address other reviewers' comments in the final version of the paper.